# In Vitro and In Vivo Test Methods for the Evaluation of Gastroretentive Dosage Forms

**DOI:** 10.3390/pharmaceutics11080416

**Published:** 2019-08-16

**Authors:** Felix Schneider, Mirko Koziolek, Werner Weitschies

**Affiliations:** Department of Biopharmaceutics and Pharmaceutical Technology, Institute of Pharmacy, University of Greifswald, 17489 Greifswald, Germany

**Keywords:** gastroretention, gastroretentive dosage forms, in vitro testing, drug release, in vivo studies

## Abstract

More than 50 years ago, the first concepts for gastroretentive drug delivery systems were developed. Despite extensive research in this field, there is no single formulation concept for which reliable gastroretention has been demonstrated under different prandial conditions. Thus, gastroretention remains the holy grail of oral drug delivery. One of the major reasons for the various setbacks in this field is the lack of predictive in vitro and in vivo test methods used during preclinical development. In most cases, human gastrointestinal physiology is not properly considered, which leads to the application of inappropriate in vitro and animal models. Moreover, conditions in the stomach are often not fully understood. Important aspects such as the kinetics of fluid volumes, gastric pH or mechanical stresses have to be considered in a realistic manner, otherwise, the gastroretentive potential as well as drug release of novel formulations cannot be assessed correctly in preclinical studies. This review, therefore, highlights the most important aspects of human gastrointestinal physiology and discusses their potential implications for the evaluation of gastroretentive drug delivery systems.

## 1. Introduction

The development of concepts for the prolongation of the gastric residence time of dosage forms has been the focus of pharmaceutical scientists for more than half a century [1]. The prolonged residence of dosage forms in the stomach, so-called gastroretention, can have various therapeutic and biopharmaceutical benefits. These include improved local drug activity in the stomach, decreased fluctuations of drug concentration in the plasma, improved patient compliance due to the reduced dosing frequency or improved bioavailability for certain drugs with absorption windows in the upper small intestine [2,3,4,5]. 

The benefits of gastroretentive dosage forms (GRDFs) were nicely demonstrated by Levy [6] already back in 1976. Levy showed that for riboflavin, the oral bioavailability can be increased if gastric emptying takes place continuously and slowly. Since riboflavin exhibits an absorption window in the upper small intestine, the controlled delivery of the drug into this segment of the gastrointestinal tract leads to better absorption. In contrast, a considerable portion of the administered dose is not absorbed if the transit through the upper small intestine is fast. This effect can be critical, since the non-absorbed fraction may cause severe colonic side effects. The question of whether a drug’s performance benefits from a prolonged gastric residence can be answered by means of a simple experiment. Lewis and colleagues [7] administered 400 mg aciclovir either in the form of 2 × 200 mg immediate release tablets, an intraduodenal infusion (500 mL at constant rate over 4 h) or a solution administered in small portions over a longer period (500 mL, 10.4 mL each 5 min for 4 h). The area under the curve (AUC) of the plasma concentration–time profile for the solution was nearly twice the value of the tablet treatment and comparable to the intraduodenal infusion [7].

Despite the overwhelming potential benefits for oral drug application, the number of gastroretentive systems marketed so far remains small [8]. The lack of proper gastroretention of various promising systems, meanwhile, outweighs the high expectations of gastroretentive systems. Therefore, a re-orientation must take place. These failures can mainly be attributed to two causes. On the one hand, there is a lack of novel and innovative approaches to achieve gastroretention. The majority of research on GRDFs so far has been based on four different concepts [3,8,9,10]. The latest of these was introduced already 35 years ago [9,11]. Since then, there have been virtually no major changes regarding the approaches to achieve gastroretention.

On the other hand, there is a lack of simple in vitro methods, which allow the early evaluation of the gastroretentive properties of new approaches. Expensive and complex clinical studies with humans are still the gold standard to confirm the gastroretentive potential of novel formulations. However, it should be the aim of pharmaceutical scientists to provide meaningful in vitro methods to test gastroretention early in development in order to support formulation development. In vivo studies should only be performed with highly promising concepts. Unfortunately, in vivo studies can also have many pitfalls that need to be considered if a qualitative statement about the actual extent of gastroretention of a system shall be made. This also requires thorough background knowledge of the most relevant physiological processes in the stomach in order to be able to correctly interpret clinical data and to draw the right conclusions for further development. This review gives a detailed overview of the physiological processes relevant for gastroretentive dosage forms and describes their implications for in vitro and in vivo investigations of gastroretentive systems.

## 2. Physiological Considerations for Gastroretentive Dosage Forms

The profound comprehension of gastric physiology in humans and laboratory animals is mandatory to understand and correctly interpret the outcome of in vivo studies on GRDFs, but also to develop new approaches for GRDFs. Since the 1970s, when the first concepts for GRDFs were developed, our knowledge about gastric physiology has dramatically increased. Modern diagnostic tools such as scintigraphy, high-resolution manometry, telemetric capsule systems, and magnetic resonance imaging (MRI) have emerged and advanced. The data obtained with these technologies already provide a very comprehensive picture of the conditions inside the GI tract. Nowadays, many factors are known to have a decisive influence on drug release and the gastric transit behavior of oral dosage forms. In this section, the most important physiological aspects will be described in more detail.

### 2.1. Gastric Motility and Transit Times

In general, the biggest hurdle for gastroretentive systems are the so-called housekeeping waves [12]. These are strong contraction waves that propagate through the distal part of the stomach to ensure that even large, indigestible objects, which remained in the stomach after gastric digestion, can be emptied. However, these intense peristaltic movements of the stomach wall are limited to phase III of the inter-digestive migrating motor complex (IMMC), a cycle that repeats every 90–120 min in fasted state [13,14]. Since these contractions are able to empty even very large objects, the gastric transit time after administration in fasted state is typically not longer than 1–2 h, depending on the phase of the IMMC at intake. However, it has to be noted that the IMMC does not always start in the stomach and that there are strong interindividual variations with respect to the duration of the IMMC. Some groups reported that it can take up to 180 min until the IMMC restarts [15,16]. With this information in mind, a proper gastroretentive system would need to have inherent gastroretentive properties to exceed this time if it is administered under fasting conditions.

After the ingestion of caloric drinks and food, the conditions within the stomach change substantially [17]. Once emptied into the duodenum, certain nutrients such as glucose or fatty acids induce a feedback mechanism that suppresses the fasting motility pattern of the stomach [18,19]. Instead, the fed state motility pattern is induced which enables the proper mixing of the contents with gastric secretions to facilitate digestion and the trituration of larger particles. In the antrum, aboral contractions with intensities comparable to phase II contractions of the IMMC crush, mix, and empty the food chyme [17]. A gastroretentive system needs to withstand these stresses without any alteration of its drug release behavior and gastroretentive properties. For gastric emptying under these conditions, size actually matters. Small particles like pellets may be mixed into the chyme and can be emptied from the stomach during gastric food digestion [20,21]. On the contrary, large and indigestible systems that are administered postprandial have a high chance of remaining in the stomach until the end of gastric chyme emptying [20,21,22]. The opening diameter of the pylorus does not exceed several millimeters and, therefore, large objects are retained until more than 90% of the ingested meal is emptied and the fasted state motility returns [22]. Therefore, any prolongation of the fed state, for instance by further meal intake, increases gastric residence time and may skew the results.

Ewe and his colleagues [19] showed that the frequent administration of smaller meals can increase the gastric residence time of non-disintegrating tablets (11 mm × 6 mm) for up to 10 h. This appears critical in view of the fact that many patients, maintaining their normal eating habits, will likely not reach fasted state during the day, as was recently demonstrated in a SmartPill^®^ study [23]. Interestingly, under fasting conditions, the gastric transit behavior of large and small objects is reversed. Small objects such as pellets are most likely trapped in the rugae of the stomach and are held back much longer than large, indigestible dosage forms [24,25,26]. When developing a GRDF, however, it would be negligent to rely solely on the eating habits of the patients. Therefore, clever concepts must be developed to achieve reproducible gastroretention even in fasted state. The implications of these considerations for a human clinical study on gastroretentive formulations will be discussed separately in Section 4.3.2.

### 2.2. Gastric Volume and pH

In addition to motility, the medium present in the stomach plays a key role for many GRDFs. Basically, the contents of the stomach represent the medium that is available for drug dissolution and distribution. The volume of fluids available for drug dissolution processes depends on the prandial state and determines the concentration of the drug within the stomach. This can have implications for drugs acting locally in the stomach, but also for the drug concentration in the small intestine, where absorption typically takes place. The concentration of a drug in the stomach is further affected by the physicochemical properties of the medium. In this regard, the pH value is probably of highest importance, especially for drugs with pH-dependent solubility. Moreover, the pH value of the gastric contents is also the basis for various formulation concepts, in which pharmaceutical scientists try to make use of the acidic pH conditions in order to obtain gastroretentive properties, for instance by the generation of gas [2,3]. Therefore, a proper knowledge of gastric volumes and gastric pH values is required for the design and evaluation of GRDFs. 

In the fasted state (i.e., after a fasting period of at least 8 h), the stomach is considered to be “empty”. In fact, it typically contains about 10–50 mL of acidic contents (pH 1–2) [27,28,29]. The administration of a dosage form together with 240 mL of water results in an increase of the volume, but also in an initial increase in pH to values of up to pH 4.6 [30,31]. Subsequently, the pH drops rapidly and the liquid is emptied from the stomach relatively fast within about 30 min (Figure 1) [27].

The intake of a caloric meal also results in higher gastric content volumes and increased pH values, but, in this case, the kinetics of subsequent volume and pH changes are clearly different from fasted state conditions. Compared to the intake of a glass of water, it can take up to several hours until volume and pH return to values observed before meal intake.

The kinetics mainly depend on the caloric value and the composition of the meal. It has been demonstrated in various studies that the emptying rate of caloric food components is significantly extended and typically accounts to 2–4 kcal/min (Figure 2) [32,33]. The initial pH value can be similar to the initial pH in fasted state, but it is usually elevated for longer times, as various components present within the food (e.g., amino acids, fatty acids) can buffer the gastric secretions [17,23]. Moreover, the viscosity and, thus, the time until a homogeneous drug distribution in the chyme is achieved may also be significantly increased [34,35].

In the case of the high-caloric, high-fat standard meal recommended by the FDA and EMA for food-effect studies and fed-state bioequivalence studies, the pH value drops from about pH 4.6 to baseline values of pH 1 almost linearly within 4 h (Figure 3) [23]. Similar results were obtained for an equicaloric liquid meal [36]. Nonetheless, the pH value in the stomach can fluctuate strongly under fasted and fed conditions [23,30,31]. These fluctuations should not affect the functionality of a GRDF. In the case of foods rich in fats, a secretion layer can be formed on top of the chyme, since the cells secreting HCl are mainly located in the proximal part of the stomach, while at the same time, mixing is rather poor [17,37,38,39]. This poor mixing in the fundus also affects the appearance of drug concentrations in plasma, as shown by Weitschies and colleagues [34,35]. They observed that a dosage form administered after the American breakfast can deposit on the chyme and may release the drug locally. Consequently, the emptying of the drug and, thus, its absorption, can be clearly delayed and may only occur after several hours [34,35].

## 3. Formulation Strategies

Over the last decades, several gastroretentive drug delivery systems were developed to overcome the physiologic hurdles present in the stomach. The majority of these systems can be assigned to at least one of the following approaches: flotation, sedimentation, expansion or mucoadhesion (Figure 4). The following section will provide a broad overview of these concepts. For a more detailed description of these concepts, we would like to refer to another review that was published in the same Special Issue of *Pharmaceutics*.

### 3.1. Floating

The very first steps towards gastroretention were made with floating systems [40]. This approach is based on the assumption that an object of sufficiently low density is able to float on top of the gastric contents. Thereby, it shall be able to evade the antral region, in which propulsive contraction waves occur [2,9]. At first sight, this approach seems reasonable, but considering the volume and composition of the gastric content, it is clear that especially under fasting conditions, the probability of a GRDF remaining in the more proximal parts of the stomach is rather low. Notably, even after meal intake, the floatation may be disturbed by factors such as meal viscosity and changes in posture [41,42].

### 3.2. Sedimentation

Almost 90 years ago, Hoelzel [43] already raised the question of whether very dense materials move slower through the gastrointestinal tract compared to less dense materials. At that time, the appearance of glass beads and silver wire was investigated in the stool after previous oral ingestion [43]. Nowadays, the analytical methods have significantly changed, but the former question is still the subject of different studies. Interestingly, the results of these studies are somewhat contradictory [24,25,44,45]. The bottom line is: the density that can be achieved technically in oral dosage forms is most likely not sufficient to cause a significant change in the gastric residence time [9]. The idea of a rapid deposition in the sinus of the stomach is further compromised by the fact that the geometry of the stomach is not the same from patient to patient and a distinct sinus region may sometimes not be present [46].

### 3.3. Expansion

The rapid expansion of a drug delivery system is often regarded as one of the most promising concepts to achieve proper gastroretention [12,47,48]. To allow a painless and safe ingestion, these systems have to increase significantly in size once they are located inside the stomach. Therefore, polymers with extensive swelling properties are often used to produce this kind of GRDF [8,49]. Another approach is to design a system that unfolds within the stomach due to the elastic properties of the used materials [8,49]. Some authors assume that an increase to a size larger than the pyloric resting diameter, which is about 13 mm, is the minimum to prevent an object from being emptied from the stomach [3,8,9]. Although this approach seems reasonable, it does not work in the fasted state as even larger objects were found to be expelled into the duodenum [48,50,51]. However, in the fed state it is indeed possible to extend the gastric residence time to many hours. This effect is due to the lack of housekeeper waves. Anyway, a proper gastroretentive system ideally is independent from the prandial status and the lifestyle of the patient. A gastroretentive system that only works reliably in patients who eat four meals, each at best with a caloric value of 800 kcal, evenly distributed over the day, is questionable, not to mention the macrosocial problems that would arise from such a lifestyle.

### 3.4. Mucoadhesion

Prolonging the gastric residence time with the aid of mucoadhesive systems is the most recent approach, which was developed already 35 years ago [11]. The idea is to extend the time of contact of the dosage form with the mucosa in combination with controlled drug release [2,8,9]. However, the use of bioadhesive/mucoadhesive systems designed to prolong the gastric residence time has not yet led to satisfactory results in humans [9,12]. Obviously, factors such as the turnover rate of the mucus, gastric secretions, as well as gastric shear stresses seem to play a bigger role than expected.

Examples in the literature show that although our knowledge in the field of gastrointestinal physiology markedly increased, the outcome of the studies performed with novel concepts of GRDFs is somewhat disillusioning. The fact that there is still not a single gastroretentive system on the market that works reliably and independently from the prandial status suggests that the approaches followed so far are not able to overcome the complex gastric environment. However, besides the concepts mentioned above, the first steps towards more innovative solutions have been made. For instance, Verma and colleagues [52] recently developed a several decimeter-long tubular device that can be inserted via the nasogastric route (Figure 5). The device shall remain in the stomach for weeks, thereby constantly releasing different drugs. After drug release is completed, it can be retrieved with the help of a magnet [52].

Of course, this system cannot be administered without the help of medical staff, but it nicely illustrates the awareness that more sophisticated approaches are needed in the future to reach the goal of reproducible gastroretention.

## 4. Characterization of Gastroretentive Dosage Forms

Currently, clinical studies with humans are still the gold standard for the investigation of the performance of new gastroretentive systems. Due to the high costs associated with in vivo studies as well due to the various pitfalls that may affect the interpretation of in vivo data, powerful in vitro methods are needed for the early evaluation of the behavior of new gastroretentive concepts. The following sections give an overview of the most frequently used in vitro methods and shall enable the reader to critically evaluate data obtained with these methods. 

The different in vitro and in vivo methods used to characterize GRDFs can be roughly divided into three groups depending on the investigated parameter:In vitro assessment of drug release behavior;In vitro and ex vivo assessment of gastroretentive properties;In vivo assessment of gastroretentive properties.

### 4.1. In Vitro Assessment of Drug Release Behavior

One of the most widely studied parameters of GRDFs is their drug release behavior, preferably under biorelevant conditions. However, the poor predictability in terms of later in vivo drug release is one of the most neglected facts in this context. Interestingly, the vast majority of scientists actually rely on compendial dissolution test apparatuses, which are usually applied for the investigation of drug release from common oral dosage forms. However, it is generally accepted that gastric conditions strongly influence the drug release behavior even for common oral dosage forms. Considering the physiological situation in the human stomach, the compendial methods are at best applicable for quality control purposes, but generally not suitable to study drug release of GRDFs [53,54]. Nonetheless, devices such as the rotating basket or the paddle apparatus as well as simple modifications of these tools are widely applied for the testing of GRDFs [55,56,57]. It is clear that the outcome of such tests largely depends on the type of investigated GRDF. 

In particular, for low-density and, thus, floating GRDFs, dissolution testing in compendial apparatuses can be problematic, as a relatively large surface area of the system is not in contact with the dissolution medium [55]. Several approaches have been proposed to impede the flotation of oral dosage forms during dissolution testing (Figure 6). The easiest and most widely accepted procedure is to use helical wire sinkers. However, when investigating a floating and at the same time expanding system, the use of a sinker may impede the swelling behavior of the dosage form [55]. An alternative is to keep the floating system underneath a ring mesh. Pillay et al. [58] and Durig et al. [59] each proposed a setup, in which they introduced one and two stainless-steel ring meshes into a paddle apparatus, respectively [58,59]. The aim of these studies was to generate more reproducible data. However, these methods are far away from the physiological conditions.

Another very simple approach was recently demonstrated by Kong et al. [60] who used a shaker incubator at a temperature of 37 °C and 100 rpm. At defined time points, samples were taken and medium was changed.

Eberle and colleagues [61] proposed a very similar approach and called it a “custom-built stomach model” (Figure 7). With this device, they tried to avoid the sticking of the dosage form to the paddle shaft and also its continuous contact with air [61]. It basically consists of Erlenmeyer flasks, which are filled with 400 mL of medium and are fixed to the carriage of a water-bath shaker [61]. Overall, this setup led to clearly accelerated drug release of a tested floating system compared to a simple USP II paddle apparatus setup. However, it must be considered that, in the paddle apparatus, the dosage form was able to float freely at the surface of the medium. In this region, shear stresses are low, which gives a good explanation for the obtained results [62,63]. The artificial nature of this setup may allow more reproducible measurement but, most likely, it will not reflect the in vivo behavior due to the lack of physiological relevance.

Parikh and colleagues [64] identified the pH differences along the gastrointestinal tract as one of the major issues that should be addressed in dissolution testing of GRDFs, especially those containing weak basic drugs [55,64]. Based on the Rosett–Rice apparatus, they developed a multi-compartmental transfer model consisting of a gastric, an intestinal and an absorption compartment (Figure 8). In this device, the drug is transferred freely from the gastric to the intestinal compartment, whereas the absorption and the intestinal compartment are separated by a filter membrane. The pH of the media can be adjusted via reservoirs containing either 1 N HCl or borate buffer. Potential benefits of this in vitro system were demonstrated for a controlled release floating system compared to an immediate release tablet [55,64]. However, it is likely that the identical in vitro dissolution and absorption profile may have also been observed for a non-floating controlled release system.

In contrast to floating dosage forms, drug release testing of mucoadhesive systems using compendial dissolution test methods seems to be less complicated. Therefore, in many cases, the paddle apparatus and rotating basket apparatus are used for this purpose. However, since mucoadhesive systems are designed to stick to the gastric mucosa, it would be misleading to study drug release if the dissolution medium is in full contact with the entire surface. Consequently, drug release may be overestimated. In a study by Llabot and colleagues [65], cyanoacrylate glue was used to fix one side of the tested monolithic system to a metal disk during dissolution experiments.

A step towards a more physiological dissolution test of novel GRDFs was made by Nakagawa and colleagues [66]. They used a modified version of the USP II paddle apparatus that was proposed by Aoki et al. [67,68] in 1992 [66,67,68]. In this model, the dissolution vessels of the paddle apparatus are filled with polystyrol beads in order to mimic physiological stresses that occur in the stomach (Figure 9) [67,68]. Nakagawa et al. [66] applied this device to a novel GRDF and demonstrated that one of the tested formulations did not provide controlled drug release due to the lack of mechanical robustness. 

A similar observation was recently observed through the use of the so-called dissolution stress test device [69]. This device was developed by Garbacz and Weitschies [70] and uses an inflatable balloon to exert realistic pressures on dosage forms (Figure 10). By implementing postprandial SmartPill data, we were able to simulate complete gastric pressure profiles as they occur in vivo. After testing the drug release behavior of marketed gastroretentive dosage forms, it could be seen that none of the systems were able to withstand these stresses [69]. These data showed that significant pressure sensitivity is one of the major issues to consider during the development of novel GRDFs. Besides drug release behavior, gastroretentive properties are endangered by intragastric stresses, even under postprandial conditions.

Besides the dissolution stress test device, there have been recent developments towards a more physiologically relevant testing of dosage forms. Among the most prominent in vitro models to simulate gastric physiology are TNO’s TIM-1 system, the human gastric simulator (HGS), and the dynamic gastric model (DGM) [53,54,71,72,73,74]. However, none of them have yet been used for drug release testing of GRDFs, but most of them offer the potential to characterize such systems in a biorelevant way.

### 4.2. In Vitro and Ex Vivo Assessment of Gastroretentive Properties

The aim of in vitro investigations on the gastroretentive properties of GRDF is to predict the actual gastric transit behavior and, thus, to identify potentially gastroretentive formulations during formulation development. These investigations are usually based on the gastroretentive concepts presented above, i.e., floating, mucoadhesive or expanding. Therefore, they typically focus on certain parameters that are believed to play a key role in vivo later. Despite various efforts in the field of gastroretentive formulation, the in vitro test methods for the assessment of gastroretentive properties have barely advanced and are still almost where they were 40 years ago. However, the development of models dedicated to the investigation of gastroretention would be very helpful for the design of novel formulation approaches.

#### 4.2.1. Floating Dosage Forms

Although simple parameters like tablet density or porosity can be used to obtain a first impression of the floating properties, these tests do not account for the dynamic changes that occur within the human gastrointestinal tract. Processes such as swelling and erosion as well as dynamically changing media may alter the in vivo floating properties of low-density systems over time. Therefore, more sophisticated methods are necessary to characterize such formulations in a physiologically relevant manner.

One step towards a more thorough characterization of the floating properties is the determination of the floating lag time and total floating time (also floating duration) [1,75]. The latter was already performed by Sheth and Toussonian [76] in 1984 for the characterization of the first marketed gastroretentive system. They used 1 N hydrochloric acid in a simple glass beaker to demonstrate the floating properties of their novel hydrodynamically balanced system (HBS) in comparison to a control capsule, which sank to the bottom [76]. The floating duration is then defined as the time that an object can remain buoyant at the surface of the test medium, determined via visual inspection. For floating monoliths, this test can be performed without any major problems. However, in the case of multi-particulates, it may occur that a portion of the tested particles sink to the bottom of the vessel or beaker after a certain time. Therefore, the number of multi-particulates that still float at the surface is usually determined at specific time points. Several research groups have removed the settled part, as well as the floating layer, and dried and weighed the particles to obtain an impression of the floating properties [55,77,78,79]. Ichikawa and colleagues [80] addressed the same issue by taking photos of the surface at certain time points and by counting the floating multi-particulates. Shah and colleagues [55], on the contrary, completely removed only the settled multi-particulates after defined time points and dried and weighed them in order to make a statement about the remaining floating multi-particulates.

In contrast to low-density systems with inherent floating properties, gas generating systems may exhibit a short time of non-flotation directly after submersion, due to the more or less slow starting reaction [55]. This so-called floating lag time may strongly influence the transit behavior, in particular under fasted state conditions, and should be as short as possible [3,75]. The floating lag time can be described as the time that is necessary for a low-density system to rise to the top of the test medium after its previous submersion [61]. The test is usually performed in an aqueous test medium that should reflect the gastric conditions [61,81]. In case of multi-particulates or raft forming systems, it may be wise to submerge the formulation within a container. For instance, Rajinikanth et al. [82] carefully submerged Petri dishes that contained a liquid raft forming system and stopped the time until flotation occurred.

The tests described above are easy to perform but it should be considered that the tested system may more or less interact with the test medium, which in turn may result in reduced buoyancy force. Already 30 years ago, Timmermans and colleagues [75] proposed another method to characterize the floating properties of a GRDF over time (Figure 11).

They used a balance to measure the resultant weight, which is necessary to keep the dosage form fully submerged [75]. Nowadays, a texture analyzer is usually applied for this test setup, but the concept remains the same. Timmermans and colleagues [75] confirmed that the swelling of a polymeric floating matrix system may lead to decreasing buoyancy force over time. Furthermore, they observed that food components may strongly affect the floating properties. 

In vivo, a clearly changed situation is expected compared to the commonly applied in vitro conditions. Besides a dynamically changing pH value, viscosity is certainly one of the major factors that has an impact on floating. So far, it has been largely neglected. In addition, the gastric emptying of the chyme and, thus, the position of the dosage form within the stomach play major roles. The time by which digestion and emptying are almost complete is certainly the most critical time for the functionality of floating systems, as by this time, the volume for flotation and the distance to the pylorus are clearly reduced. Moreover, the dosage form will be exposed to higher stresses if it is located in the distal part of the stomach. The mentioned tests may be useful to get a first impression of the floating properties but in vivo, floating depends on far more complex processes.

#### 4.2.2. Sinking Dosage Forms

In general, the same factors that are important for the performance of floating dosage forms are also relevant for sinking systems. These include the viscosity of the medium or changes in the total density due to the presence of swelling. Since the number of studies on sinking systems is extremely limited, data on characterizing their gastroretentive properties in vitro are lacking too. However, compared to floating dosage forms, the performed tests seem somewhat easier. Among these are the theoretical calculation of the density, the density determination via air comparison pycnometry, and the visual determination of the settling kinetics [24,44,77,83].

#### 4.2.3. Expanding Dosage Forms

The simplest way to investigate expanding dosage forms is to perform a measurement of the system before and after expansion. For instance, Gröning and colleagues [84] measured their expandable collagen sponges with predefined dimensions before and after swelling in different media. Dorozynski [85] characterized the extent of the increase in size of their gastroretentive capsule by measuring the Feret’s diameter and perimeter. 

Furthermore, they investigated the water uptake (also swelling index or swelling ratio) of their formulations, which is the most widely performed test for such systems [85,86,87,88]. After full submersion into a certain fluid, the dosage form is removed from the test medium at defined time points and weighed. With knowledge of the initial mass (before swelling), the swelling index (SI) can be calculated using the following equation:$$SI\left(\%\right)=100\times \;\frac{\mathrm{ms}\left(t\right)\;-\;m\left(i\right)}{m\left(i\right)}$$
where m_s_(t) refers to the mass determined for the swollen system at time t and m(i) equals the initial mass of the system. The obtained data can also be used to determine the equilibrium swelling time, i.e., the time until complete swelling and size increase [89]. Some authors further extend this test by investigating the erosion. The swollen system is then dried to a constant weight (m_d_(t)) so that the mass loss referred to the initial weight can be used to describe the extent of erosion (*E*) that took place [86]:$$E\left(\%\right)=100\times \;\frac{m\left(i\right)-\mathrm{md}\left(t\right)}{m\left(i\right)}$$

A major problem with the methods described is the stress that occurs during the removal of the formulation from the media. Similar to hydrogel matrix tablets, expanding formulations based on hydrophilic polymers can be pressure sensitive and, thus, the destruction of the gel may occur within the human GIT. However, it should be noted that the gastric contents are not described correctly by the use of aqueous media. The swelling properties of these systems in more complex media (e.g., emulsions) may be significantly different. 

One method, mainly applied to superporous hydrogels, is the investigation of the systems’ transparency. Omidian and colleagues defined the Tcore as the time at which the swelling system changes from opaque to transparent appearance [90]. However, this test can be only applied if the system performs a visual transformation and if the test media are also transparent. The investigation in an emulsion, for example, milk, would hardly be possible.

The exposed size parameter was introduced by Klausner et al. [91] who wanted to characterize their gastroretentive system in 2D in vivo X-ray imaging [91]. The exposed size parameter (ESP) is calculated by the following equation:$$ESP\left(\%\right)=100\times \;\frac{\mathrm{Ll}\times \mathrm{Ls}}{S}$$
where Ll and Ls are the distances between two markers in two directions, so that the area of a rectangle results. *S* is the area of the dosage form before folding and, therefore, describes the maximum expectable value.

This method provides a non-invasive measurement of the size increase but is mostly restricted to systems that unfold and that can sufficiently be described by only two dimensions. Although, originally developed for the in vivo assessment of the size increase, Wang et al. [92] used this parameter to describe their system in vitro.

#### 4.2.4. Mucoadhesive Dosage Forms

The gastroretentive properties of mucoadhesive systems depend mainly on the strength of the interaction between the surface of the dosage form and the gastric mucosa. Therefore, the most widely used method to determine the mucoadhesive properties of GRDFs is the measurement of the tensile strength [87,93,94,95,96,97]. Since first described, progress in the field of force measurement has substantially increased and so did the quality of the obtained data. While the first tests had to be performed with balances, nowadays, texture analyzers with appropriate load cells are mainly used. For such tests, the formulation or tested polymer is fixed at the movable probe head of the texture analyzer while the mucous tissue is fixed onto a sample holder below the probe. The tested polymer sample is then lowered at a constant speed until a defined force is measured upon contact with the tissue. Afterwards, this force is either held for a certain time or the probe is raised immediately. The maximum force that is necessary to detach the sample from the tissue is defined as detachment force [98]. Another important parameter in this context is the work of adhesion, which is displayed by the area under the force–distance curve obtained under these test conditions [93,97]. Ponchel and colleagues [97] suggested that this parameter has a greater importance than the detachment force. Nowadays, researchers often provide both values [97,99].

For optimal adhesion, the mucoadhesive formulation should obtain close contact with the tissue in the dry state [99]. However, this is rather unlikely to occur in vivo as was highlighted by Laulicht and colleagues [100], who found that detachment forces measured in vitro and in vivo are not comparable. A critical parameter for this test is certainly the initial force that is applied to ensure sufficient contact of the formulation and the tissue specimen. Values in the literature range from about 0.05 N for 5 min to about 1 N for 15 min [87,92,94]. It seems very unlikely that these forces occur in a controlled way in vivo. 

A nearly forceless assessment of the mucoadhesive properties of such systems can be done with the gut sac test [98,101,102]. For this test, a tissue specimen is submerged into a dispersion that contains a defined concentration of mucoadhesive particles. After a defined time, the sample is removed and the percentage of binding is determined. This can be done either by counting the particles adhering to the tissue or, as performed by He and colleagues [102], by counting the particles in the suspension before and after the test using a Coulter counter [102,103]. Lehr and colleagues [103] chose a comparable procedure and perfused rat intestine in situ (Figure 12). Before and after the passage of the rat intestine, microparticles were counted to assess the mucoadhesion [103]. Possible problems mainly arise from undefined agitation during the test as well as from the pre-treatment and the quality of the tissue. It was shown that the mucous thickness can vary significantly among different animal species and even within the same subject along the GIT [99].

Another widely accepted test to assess the mucoadhesive properties of multi-particulates is the wash-off test [98,104,105]. The general preparation steps for this test are largely identical among research groups. These include dissection of mucous tissue from freshly slaughtered animals (e.g., rat or pig) and the cleaning of the tissue and its cutting into pieces of defined dimensions. Afterwards, the tissue is loaded with the potential mucoadhesive formulation and fixed at a predefined angle (about 40–50°) [104,105]. The multi-particulates are then rinsed off the tissue. Either the amount of remaining multi-particulates adhering to the tissue or the time to detach all of the formulation from the tissue can be measured [104,105].

A comparable setup was described by Prajapati and colleagues [106] who used the disintegration tester to perform this test. The tissue was tied to a glass slide and fixed to a disintegration tester. After a defined number of reciprocating movements, the test is stopped to determine the quantity of multi-particulates still adhering to the tissue. 

Compared to the ex vivo situation, the in vivo adhesion of GRDFs to gastric mucosa is likely to be reduced because of high turnover rates of the mucus, especially after food intake [12]. It is not difficult to imagine that a mucoadhesive system is easily sloughed off from the mucosa at a secretion rate of up to 10 mL/min [37]. Moreover, the shear stress transferred by gastric contraction waves via the chyme is also often not considered. It can be assumed that the controlled attachment to the fundic region, where mixing is rather poor, holds the highest potential for mucoadhesive systems to withstand the gastric environment. However, literature indicates that the oft-stated adhesive interactions are not specific for mucous tissue but also work on other surfaces [107]. In many cases, a prolonged residence time, especially under fasted conditions, rather seems to be the result of the size. A small particle size leads to a high probability of entrapment in the mucus and in the gastric rugae [24,25,26]. Thus, it is not an effect of explicit mucoadhesion that leads to longer gastric residence time.

#### 4.2.5. General Models

In a review by Omidian and colleagues [90], they described the mechanical testing of their superporous hydrogels using a novel gastric simulator. Although a thorough description of this device was missing, it showed the awareness that the mechanical robustness of novel GRDFs is of major importance for their success in vivo [90].

Chen and colleagues [108] developed a novel in vitro model with special focus on the anatomical appearance of the stomach (Figure 13). With the use of three strings they were able to generate local contractions to mimic the antral region [108]. The dimensions and also the gastric rugae were correctly simulated using an actual human stomach as a template [108]. Although this device was designed for the simulation of digestive processes, the model can potentially be used for the investigation of gastroretentive systems.

A novel approach was also chosen by Neumann and colleagues [47], who tried not to align their test method to the type of GRDF, but developed a model which simulates housekeeper waves in vitro (Figure 14) [12,47]. Using a simplified antral model, they were able to investigate the influence of different material properties on the emptying behavior of objects. By this, they could disprove the hypothesis that a GRDF has to have a slippery surface to evade antral contraction waves [48,109]. This was actually the property that ensured rapid emptying of the tested objects in vitro [47].

In general, it is complicated to simulate the complex conditions that are present in the human stomach. Although various approaches over the last years are heading in the right direction, it is still not possible to correctly predict the actual transit behavior of a gastroretentive system. This is also due to the fact that we are not yet aware of the exact properties necessary to reproducibly increase gastric residence time.

### 4.3. In Vivo Assessment of Gastroretentive Properties

It should be clear by now that the current in vitro methods are, at best, useful only for initial statements about the gastroretentive properties of new formulations in terms of the respective principle used to provide gastroretention. However, in most cases, the in vivo transit behavior cannot be predicted. Therefore, in vivo investigation remains the only possibility of checking whether a supposedly gastroretentive formulation actually exhibits the desired transit behavior or not.

#### 4.3.1. Considerations for Animal Studies on Gastroretentive Dosage Forms

During the early development of novel dosage forms, animal studies are widely used and often not replaceable. For first tests regarding the functionality of a gastroretentive concept, especially multi-particulate mucoadhesive or floating systems, smaller species such as rats or rabbits are used [110]. Although the rabbit is one rare example of an animal model that exhibits gastric pH values comparable to humans, it is not hard to imagine that transit data obtained from the rabbit’s stomach, especially for larger dosage forms, can hardly be transferred to humans. 

In order to administer dosage forms that were initially developed for humans (e.g., tablets or capsules), larger species like dogs and pigs are used. In this respect, the beagle dog is still the standard [111]. However, recent publications demonstrated that the data obtained with this animal model has to be treated carefully. Although, several physiological parameters are comparable to humans, the intragastric stresses, especially, as well as the gastric transit time of objects are considerably higher in these dogs [112,113,114]. The recorded pressures in dogs during gastric emptying can amount to twice the values of humans [113,115]. Moreover, dosage forms are exposed considerably longer to such intragastric stresses. Although, the fasted-state transit of large objects is comparatively high but more or less in the range of the transit in humans, the situation is clearly changed after postprandial intake [114]. A previous meal of 10 g already increases the gastric residence time of a telemetric capsule up to 9.4 h [114]. The same trend was observed in other studies [116,117]. One explanation could be the different onset of the MMC in dogs. Although, the duration of the inter-digestive motility cycle is comparable to that of humans [116], its occurrence after feeding requires much more time than in humans [118]. The mentioned differences also affect the transit behavior of gastroretentive systems and promising results from dog studies are hardly transferrable to humans. For instance, Cargill and colleagues [119] obtained a gastric residence time for their system of more than 24 h in the dog but could not reproduce these data in humans, where residence times of about 6.5 h were observed [51,119].

The pig is often claimed to be an alternative to the dog, especially because of the length of the small intestine as well as the comparable microbiota of the colon with respect to humans [120]. A recent example of the use of pigs to investigate the in vivo behavior of gastroretentive dosage forms is a work by the group led by Robert Langer [121] at the Massachusetts Institute of Technology in Boston. They demonstrated gastric residence times of their novel system of more than one week in Yorkshire pigs [121]. The authors concluded that the system can also be used successfully in humans [121]. However, nearly 30 years ago, Hossain and colleagues [122] reported that common larger, non-disintegrating dosage forms remained in the stomach of pigs for several days. To obtain comparable data in humans is unimaginable. Aoyagi and colleagues [116] made similar observations and raised the question if pigs may not have a distinct inter-digestive motility comparable to humans. Another explanation could be the more restrictive opening diameter of the pig’s pylorus [116]. This hypothesis is strengthened by Davis and colleagues [123], who observed the fasted-state gastric residence time for pellets of 2.2 h. Larger tablets even remained for up to 6 h in the pig stomach after fasted-state administration [123]. An extreme example for a poor animal model with respect to the investigation of the gastroretentive potential is the dairy calve. In a recent study, we found gastric residence times for telemetric capsules that exceeded the power supply of the capsules (>5 days) [124]. Such examples demonstrate that comparable dimensions of the gastrointestinal segments are not necessarily indicative of the comparable transit behavior of dosage forms. This has to be kept in mind, especially, during the investigation of gastroretentive dosage forms. In our opinion, the functionality of a novel gastroretentive system can only be confirmed in vivo by conducting a well-planned human study.

#### 4.3.2. Considerations for Human Studies on Gastroretentive Dosage Forms

A study in human subjects remains the gold standard. However, if not performed with the critical pitfalls in mind, this cost-intensive type of study may also result in misleading data. Therefore, the major issues that must be taken into consideration will be addressed in the following sections.

##### To Feed or Not to Feed?

The greatest influence on gastric residence time is certainly the intake of caloric food components. As was described above, the greatest risk of early gastric emptying is present in fasted state due to the action of housekeeper waves [12]. Since no type of gastroretentive formulation is inherently gastroretentive, in most cases it will be necessary to ingest at least a small meal previous to dosage form administration. The significance of this thought becomes clear when thinking of expandable systems. An early gastric emptying with subsequent expansion in the small intestine can lead to severe consequences for the patient (e.g., intestinal obstruction). Attention should be paid to the amount and composition of the meal. Depending on the caloric content, the transit data can vary considerably. Berner and Cowles [125], for example, already observed a significant reduction in the transit time of their gastroretentive formulation after reducing the fat content of a co-administered meal from 50% to 30%. According to Waterman [12], there is a linear relationship between gastric residence time and caloric content of a previously given meal (Figure 15). 

When testing GRDFs, a good recommendation is to administer a light meal of up to 250 kcal in order to keep the influence of the food on the prolongation of the gastric transit time as low as possible. The gastric residence time then strongly depends on the onset of fasted state motility, especially IMMC phase III. It is possible to estimate the expectable gastric residence time (EGRT) of a non-gastroretentive dosage form via the following rule of thumb:$$\mathrm{EGRT}\left(\min\right)\approx \frac{\mathrm{ingested}\;\mathrm{calories}\;\left(\mathrm{kcal}\right)}{\mathrm{gastric}\;\mathrm{emptying}\;\mathrm{rate}\;\left(\frac{\mathrm{kcal}}{\min}\right)}+\mathrm{time}\;\mathrm{to}\;\mathrm{IMMC}\;\mathrm{phase}\;\mathrm{III}\;\left(\min\right)$$
$$\mathrm{EGRT}\left(\min\right)\approx \frac{\mathrm{ingested}\;\mathrm{calories}\;\left(\mathrm{kcal}\right)}{2-4\;\frac{\mathrm{kcal}}{\min}}+0-180\;\min$$

Based on the above stated rule of thumb, the postprandial intake of a large, non-disintegrating dosage form after the ingestion of the American breakfast (approximately 900 kcal) would lead to expected gastric residence times of 3.75–10.5 h. Of course, these values can be considered as rare extremes. However, when one looks at the graph in Figure 15 based on these calculations, it is clear that the calculations made by Waterman fit perfectly.

On the basis of the recommendation to administer a 250 kcal meal, the maximum expectable gastric residence time for a larger, non-disintegrating dosage form would be approximately 5 h. A true GRDF should clearly exceed this time. Further food intake may artificially prolong the postprandial status of the subjects and should be avoided until the desired gastric residence time is reached or the system is emptied from the stomach [19]. In general, the influence of the co-administered food should be as small as possible but should allow enough time for the system to exhibit its gastroretentive properties (e.g., due to the presence of swelling or gas generation). The same thought is also true for the administration of non-caloric liquids. Their frequent administration would artificially prolong the gastric residence of floating systems. On the other hand, the co-administered liquid has to be sufficient in order to assure gastric arrival. It is known that orally administered dosage forms may be retained in the esophagus for hours, unnoticed by patients [17]. Thus, not only rapid gastric emptying into the duodenum but also esophageal retention must be kept in mind when GRDFs are to be tested in vivo.

##### Choosing the Right Control Formulation

There are several examples in the literature where choosing none or the wrong control formulation has led to a misinterpretation of the in vivo data [12,126,127,128]. The most important parameter to mention is the drug release behavior. It makes no sense to test a slow releasing GRDF against an immediate release formulation and then to conclude a gastroretention based on the prolonged plasma or urine excretion profile [12,126,127,128]. The in vitro release behavior of the reference formulation should be ideally the same or at least comparable to the test formulation. This also applies to the expected drug release in vivo. As already mentioned, it is hardly possible to predict the exact in vivo release profile, but the in vitro characterization should be as extensive as possible. Besides biorelevant media, it is important to focus on mechanical stresses during in vitro testing. A hydrogel matrix tablet that releases the drug in the paddle apparatus in a manner comparable to the tested system may show dose dumping in vivo due to the gastric motility [129]. In this case, it would represent a rather poor control formulation. Besides comparable drug release behavior, the control formulation should have the same initial size and shape as well as a comparable disintegration behavior. The properties, which are believed to lead to sufficient gastroretention should of course be missing. A floating capsule could be tested against a capsule of comparable size, but without floating properties [12]. An expanding tablet could be tested against a comparable dosage form that ideally shows no change in size.

##### Choosing the Right Model Drug

In order to verify the gastroretentive properties, certain pharmacokinetic parameters such as t_max_ or AUC may be determined. However, the problem with this procedure is that an appropriate model drug has to be selected in order to be able to draw the right conclusions from the study. Theophylline, for instance, is a good example for a rather poor model drug, since it is well absorbed along the whole GIT. In this case, it is hardly possible to state whether the dosage form is still in the stomach and releases its drug locally. A model drug, at best, exhibits an absorption window in the upper GIT, such as riboflavin [6,12,48]. Another possibility would be to use substances that degrade under either gastric or intestinal conditions. To evaluate the gastroretentive properties based on pharmacokinetic data requires thorough planning of the in vivo investigation. However, to give a clear statement about the gastric residence time, such studies are ideally combined with an appropriate imaging method to visually confirm the intragastric location.

##### Choosing the Right Imaging Technique

Novel imaging techniques such as magnetic resonance imaging or scintigraphy have dramatically enhanced our possibilities regarding non-invasive dosage form visualization and in vivo tracking. The following section will briefly describe commonly applied in vivo methods used to describe the in vivo transit behavior of GRDF.

Scintigraphy is often referred to as the gold standard of dosage form tracking. A high temporal resolution in combination with sufficient spatial resolution are certainly strong arguments in favor of this technique. One of the first studies to highlight the value of scintigraphy in determining the performance of a formulation claimed gastroretentive was done by Bennett, Hardy, and Wilson [41] in 1984. In this study, the gastric emptying behavior of a ^113m^In-labeled raft-forming system was studied under varying body postures. For the first time, visual evidence was presented that floating systems are more rapidly emptied from the stomach if subjects are lying on the left side, in which case, the antral region represents the highest point [41]. However, one major drawback regarding scintigraphy is the fact that anatomical information is missing. It may be unclear whether a formulation is located within the stomach or within an intestinal segment nearby. To overcome this issue, Gansbeke and colleagues [130] chose an interesting approach when investigating a floating capsule formulation. A meal containing a radionuclide with a short half-life, in this case ^99m^Tc, was administered to delineate the contour of the stomach. A second radionuclide with a longer half-life was then used to label the dosage form [130]. Recently, Pund and colleagues [94] successfully applied scintigraphy to visualize a mucoadhesive formulation in human subjects. However, in recent years, this technique has not often been used due to the decreased availability of the necessary radionuclides and regulatory issues in connection with the radiation exposure of the subjects.

The use of X-ray for dosage form tracking raises the same issues as scintigraphy, i.e., radiation exposure to healthy subjects. Besides the missing anatomical reference, the high radiation exposure for subjects remains one of the major drawbacks of this method, which explains why this method is rarely applied. For the evaluation of whole GI transit, several X-rays have to be gathered. However, confirming intragastric location could be done with a few X-rays after a certain time. Therefore, there have been some examples that highlight the potential value in locating gastroretentive systems. Klausner and colleagues [131] successfully tracked their gastroretentive system in humans using X-ray. With the aid of 5 mm radio-opaque threads obtained from surgical gauze pads, they could even determine its unfolding behavior in vivo (Figure 16) [131].

A comparable method was used by Sharma et al. [132] to visualize barium sulfate-loaded floating tablets in rabbit stomach. Vemula and colleagues [133] applied this method to characterize the transit behavior of sustained release tablets. However, only solid dosage forms can be tracked, which also have to contain comparatively large amounts of labeling material. Depending on the type of gastroretentive system, this can lead to significant changes regarding the formulation, which has to be considered in early development. The amount of radio-opaque substances can be reduced; however, if an appropriate spatial resolution has to be achieved, the radiation dose then has to be increased [134,135]. Therefore, the use of X-ray for the localization of gastroretentive systems will likely not significantly exceed the already mentioned applications in the near future.

##### Magnetic Resonance Imaging

Magnetic resonance imaging is probably the best example for a non-invasive measurement technique that provides anatomical reference data with sufficient spatial resolution. A great advantage of this technique is the clearly reduced risk of radiation exposure of the subjects. On the other hand, the temporal resolution of classical MRI measurements is comparatively low. Since body movement during an MRI scan leads to artifacts in the images, the subjects are usually advised to hold their breath, which reduces the feasible sampling rate. However, for the characterization of dosage form transit, it seems completely sufficient to apply a sampling rate of 0.5 min^−1^. This can be further reduced if only the gastric location of the dosage form should be confirmed. 

Recently, Curley and colleagues [135] could even determine drug disintegration in vivo with this technique. Marciani et al. [136] furthermore visualized and confirmed the raft formation of an alginate containing antacid via MRI. However, for most applications, an appropriate labeling is necessary to correctly localize a dosage form within the GIT. Steingoetter and colleagues [42], for example, used black iron oxide to characterize the floating behavior of a gastroretentive tablet formulation via MRI (Figure 17). 

They also accounted for the effect of body posture on the floating and emptying behavior and conducted their study in an upright 0.5 T MRI scanner [42]. Grimm and colleagues [137] recently confirmed via MRI that there is no difference in gastric emptying of floating and non-floating gastroresistant capsules administered under fasting conditions.

##### Magnetic Moment Imaging

By use of a magnetic dipole signal and appropriate sensors, it is possible to precisely track dosage forms in real-time and in three dimensions. The radiation exposure and risk of patients can be neglected in this connection. In the case of magnetic marker monitoring, extremely sensitive sensors are applied, which also require measurement inside a magnetically shielded room to reduce the noise of the Earth’s magnetic field [138,139]. About 10 mg iron oxide are sufficient to achieve high spatial and temporal resolutions [138]. The iron oxide can be directly included into the formulation or even be introduced later, e.g., into drilled holes [139]. Other methods work outside a shielded room but require stronger magnetic moments, i.e., larger permanent magnets [140,141,142]. Depending on the type of gastroretentive system and its formulation, such a modification may be impossible. In comparison to the signal spreading during scintigraphic measurement, there is signal loss due to the disintegration [138]. This circumstance has been successfully used to estimate the disintegration behavior in vivo [143]. Major drawbacks are the necessity of a sufficient magnetic dipole moment and the restriction that only one signal is detectable at a time. Therefore, this technique is only applicable to rather solid monolithic devices.

##### Gastroscopy

A more straightforward but highly invasive way to confirm the intragastric localization of a dosage form is the visual examination via an endoscope [144]. The anatomical reference is inherently given. In most cases, a flexible fiber-optic endoscope is used. Klausner and colleagues [91], for instance, confirmed the unfolding of their novel gastroretentive device by gastroscopic evaluation in dogs. However, the gastroscope usually does not remain within the stomach for extended time periods and, thus, has to be reinserted after certain intervals during the whole evaluation of gastroretentive systems. An alternative approach could be the one-time inspection after a predefined time or the application of a telemetric capsule system containing a camera. For instance, Pedersen and colleagues [145] recently used the PillCam™ to visualize different oil-filled capsules within the human stomach. A major drawback of this technique is the freely moving nature of the device. Its gastric emptying before the emptying of the gastroretentive systems form cannot be excluded, which means that the endoscopic capsule system should be administered with a significant but also sufficient delay. As an approximation, the above stated formula for estimation of the expectable gastric transit time of dosage forms can be used (Section 4.3.2). The endoscopic capsule should then be administered shortly after this time to confirm a gastric residence beyond that of common dosage forms.

##### Ultrasonography

Although ultrasonography suffers from comparatively low spatial resolution, its high temporal resolution along with the technological advancements over the past years provide a basis for potential use for localization of gastroretentive dosage forms. Novel systems are portable, wireless, and easy to use [146]. The literature on ultrasonographic measurement to investigate dosage form location within the GIT is very limited [147]. However, a few examples for its successful application exist and date back to 1988 [148,149,150]. A major problem are gases within the gastrointestinal tract that serve as a reflector and, thus, hamper the signal acquisition [138]. The small and large intestines are especially filled with gas, and the simultaneous imaging of all segments of the gastrointestinal tract remains difficult. For instance, Maublant and colleagues [150] could detect a tablet within the stomach but the signal was lost after its passing into the intestine. However, for the confirmation of intragastric location of a gastroretentive system these problems are largely neglectable. On the other hand, the disintegration behavior of the system is crucial, as was demonstrated by Amital and colleagues [149]. The detection of rapidly disintegrating tablets by ultrasonography was nearly impossible after a few minutes [149]. Therefore, intragastric imaging of dosage forms is likely restricted to slow or non-disintegrating systems [149]. Besides the disintegration behavior, a sufficient amount of surrounding medium seems critical to obtain a good contrast. Shalaby and colleagues [109] investigated their superporous hydrogel and suggested to administer a glass of water immediately before image acquisition [109]. These few but good examples indicate that, at the moment, the potential of an ultrasonographic measurement is probably underestimated when it comes to confirming the intragastric location of novel gastroretentive systems.

## 5. Conclusions

Gastroretention of dosage forms offers enormous potential. A platform technology in this area would substantially change the pharmacotherapy with orally administered drugs. However, over the past five decades, such a technology has yet to be developed. This is not only due to the lack of predictive power of the applied in vitro methods, but also the misinterpretation of in vivo data. In the development of such systems, well-known paths must be left, and less attention must be paid to the marketing of the system. With the help of novel developments in the field of diagnostics and in vitro testing, completely new possibilities are already available today. These should be applied in a smart and sophisticated way in order to reach the goal of reproducible gastroretention.

## Figures and Tables

**Figure 1 pharmaceutics-11-00416-f001:**
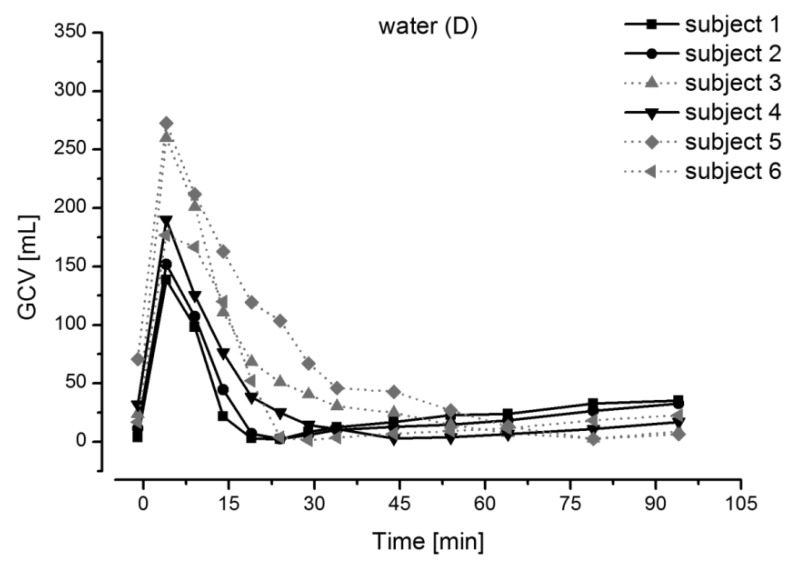
Gastric content volume (GCV) changes after administration of 240 mL of water in fasted state. Reprinted from Reference [27] with permission from the American Chemical Society.

**Figure 2 pharmaceutics-11-00416-f002:**
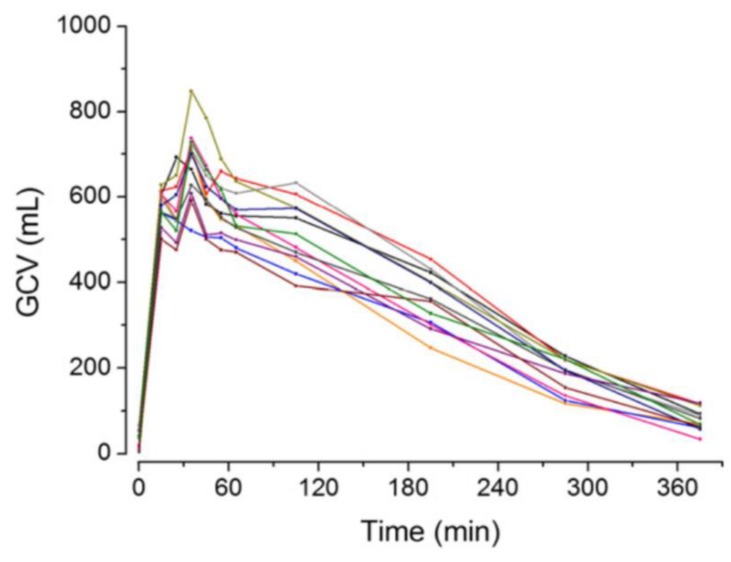
Gastric content volume (GCV) after intake of the high-caloric, high-fat standard breakfast as used for food-effect and fed bioequivalence studies. Reprinted from Reference [32] with permission from the American Chemical Society.

**Figure 3 pharmaceutics-11-00416-f003:**
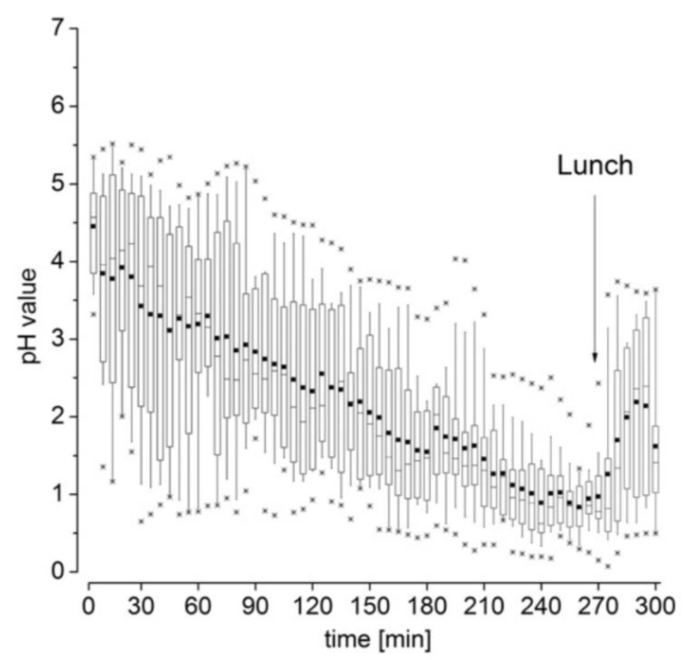
Gastric pH values after administration of the SmartPill^®^ in fed state. Reprinted from Reference [23] with permission from Elsevier.

**Figure 4 pharmaceutics-11-00416-f004:**
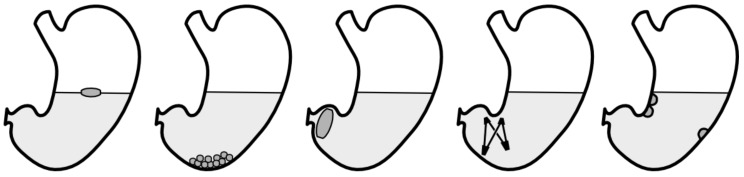
Schematic of the most common approaches to achieve gastroretention. From left to right: floating, sedimentation, expansion (swelling), expansion (unfolding), mucoadhesion.

**Figure 5 pharmaceutics-11-00416-f005:**
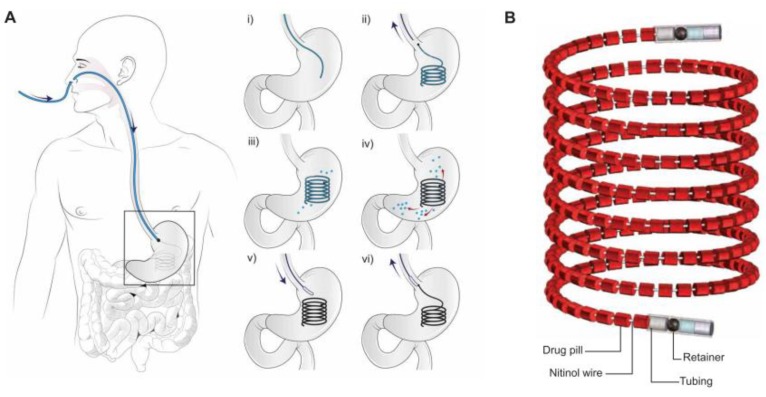
Schematic of the intragastric placement (**A**) and detailed view (**B**) of the tubular gastroretentive device proposed by Verma et al. and reprinted from Reference [52].

**Figure 6 pharmaceutics-11-00416-f006:**
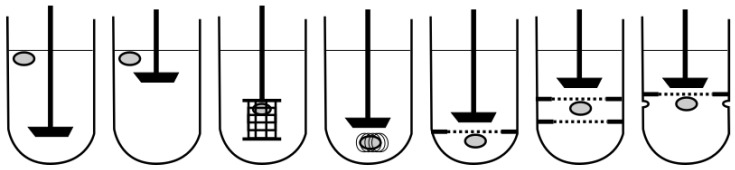
Compendial dissolution apparatus and modifications. Adapted from References [55].

**Figure 7 pharmaceutics-11-00416-f007:**
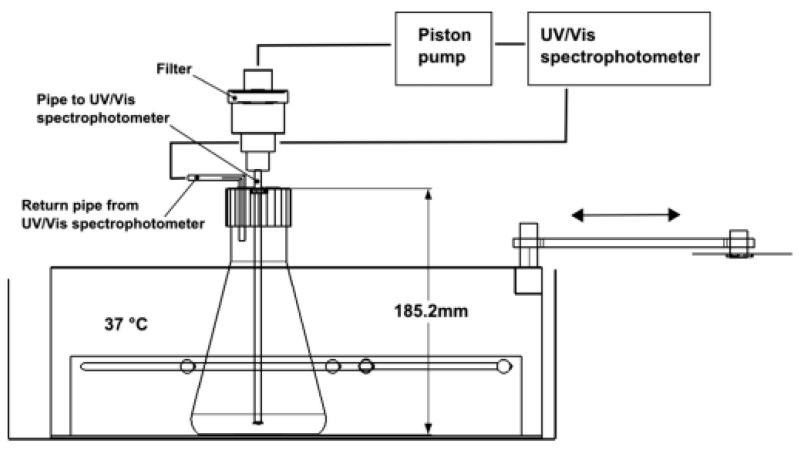
Schematic of the “custom-built stomach model”. Reprinted from Reference [61] with permission from Elsevier.

**Figure 8 pharmaceutics-11-00416-f008:**
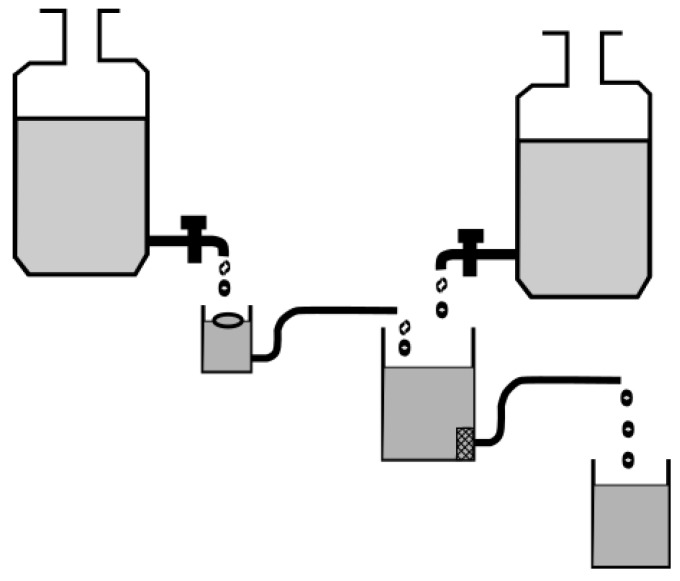
Schematic of the multicompartment dissolution apparatus. Adapted from Reference [55].

**Figure 9 pharmaceutics-11-00416-f009:**
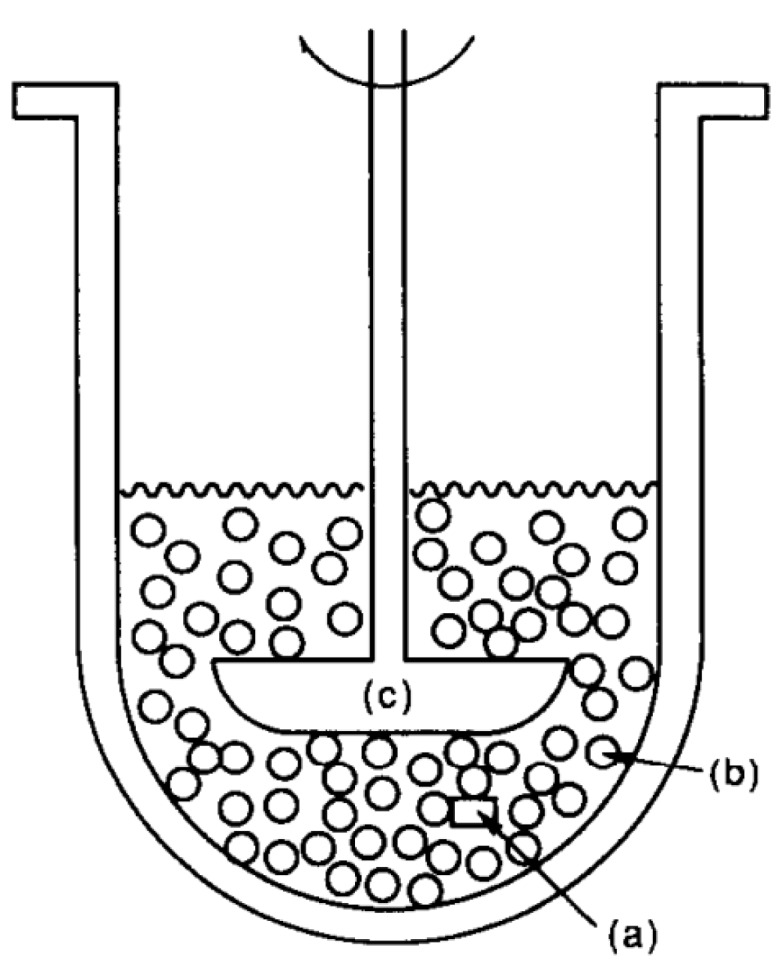
Schematic of the paddle-bead method. Reprinted from Reference [68] with permission from Elsevier. Explanation of letters as stated by the authors: (**a**) matrix tablet, (**b**) polystyrene beads, (**c**) paddle.

**Figure 10 pharmaceutics-11-00416-f010:**
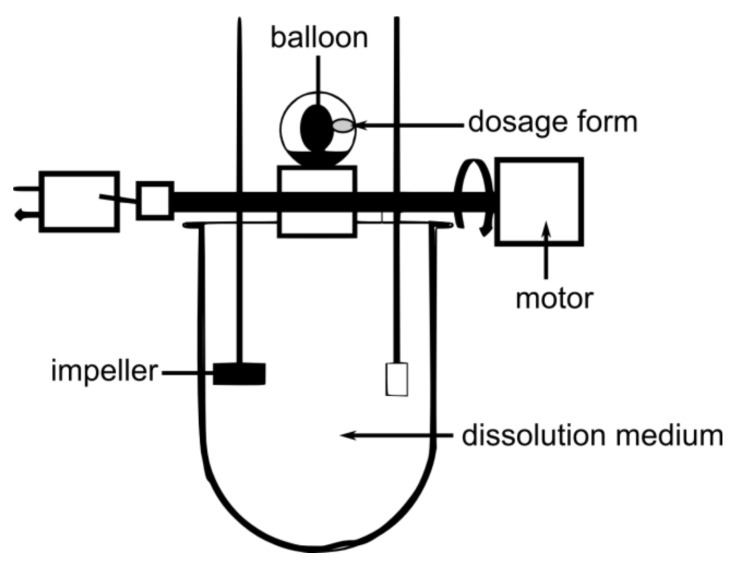
Schematic of the dissolution stress test device. Adapted from Reference [70].

**Figure 11 pharmaceutics-11-00416-f011:**
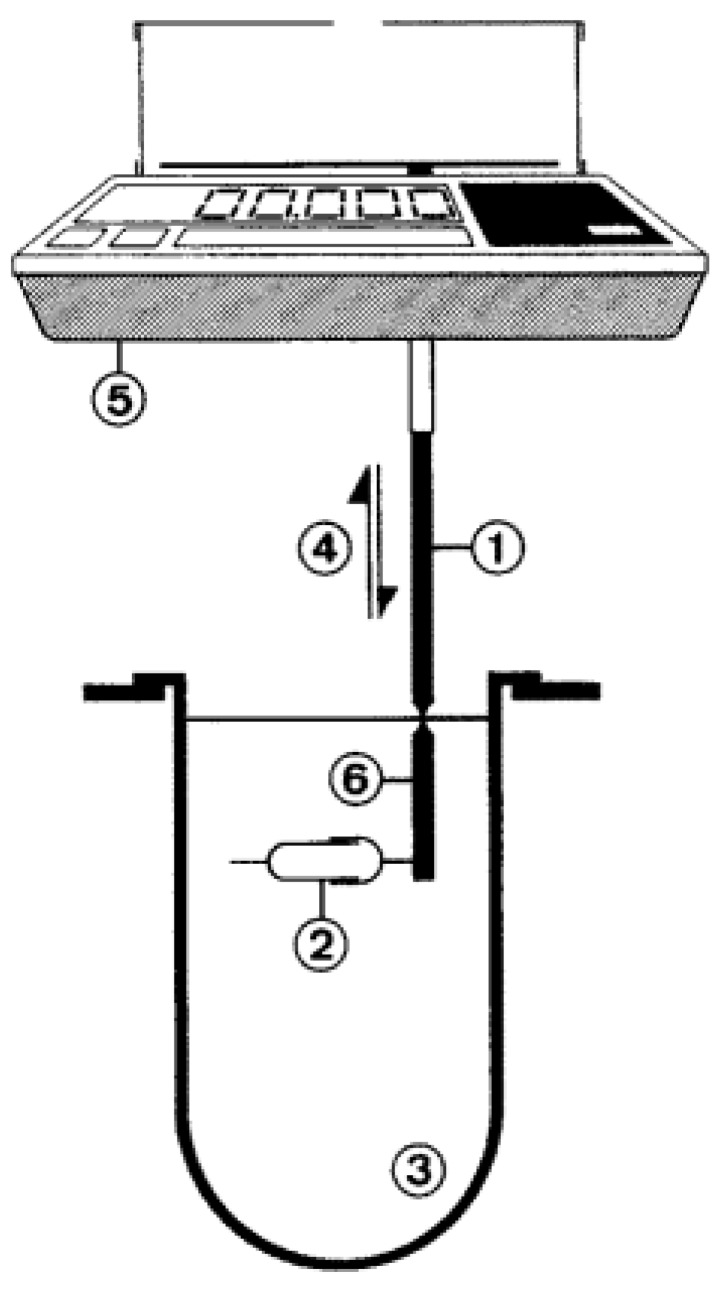
Schematic of the resultant-weight measurement proposed by Timmermans et al. and reprinted from Reference [75] with permission from Elsevier. Explanation of numbers as stated by the authors: (1) linear force transmitter device, (2) test object, (3) fluid medium, (4) direction of reacting force F, (5) weighing balance, (6) spit-holder extremity.

**Figure 12 pharmaceutics-11-00416-f012:**
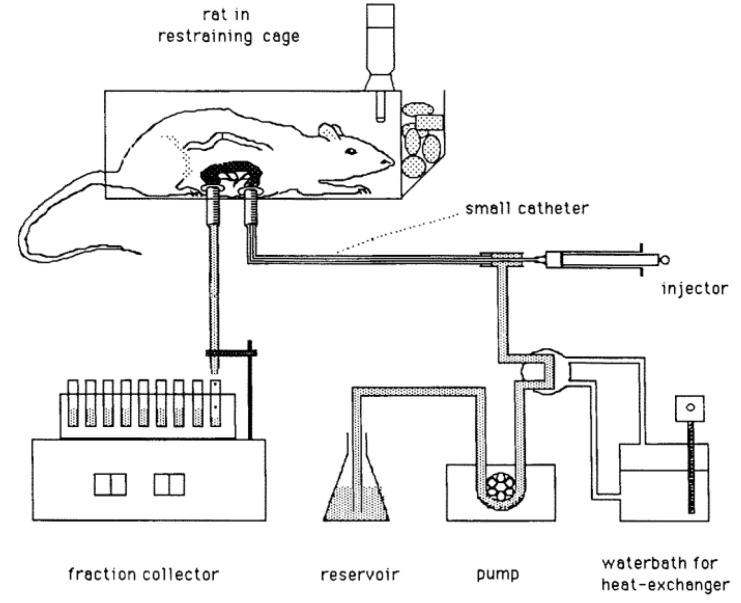
Schematic of the rat intestine perfusion test used to evaluate the mucoadhesive properties. Reprinted from Reference [103] with permission from Elsevier.

**Figure 13 pharmaceutics-11-00416-f013:**
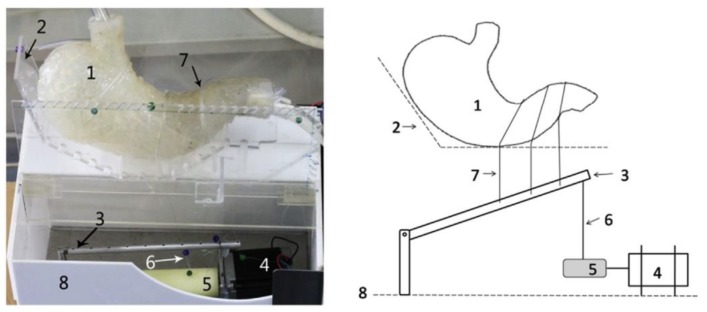
Photograph (**left**) and schematic (**right**) of the human stomach model proposed by Chen et al. and reprinted from Reference [108] with permission from Elsevier. Explanation of numbers as stated by the authors: (1) the human stomach model, (2) plexiglass support, (3) pull rod, (4) step motor, (5) spindle, (6) wire, (7) contractive ropes, (8) base box.

**Figure 14 pharmaceutics-11-00416-f014:**
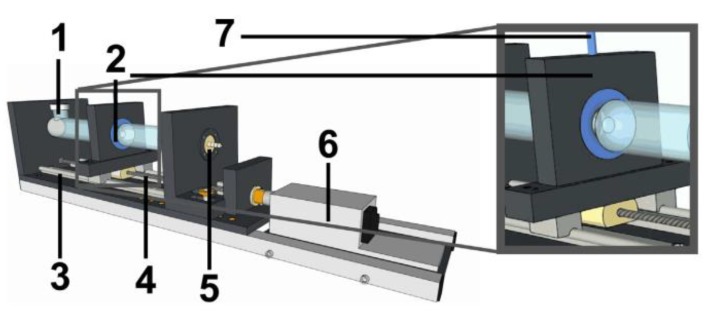
3D illustration of the mechanical antrum model proposed by Neumann et al. and reprinted from Reference [47] with permission from Elsevier. Explanation of numbers as stated by the authors: (1) sample inlet, (2) movable slide with inflatable balloon, (3) guide rails, (4) drive screw, (5) media supply via plug, (6) stepping motor, (7) compressed air supply.

**Figure 15 pharmaceutics-11-00416-f015:**
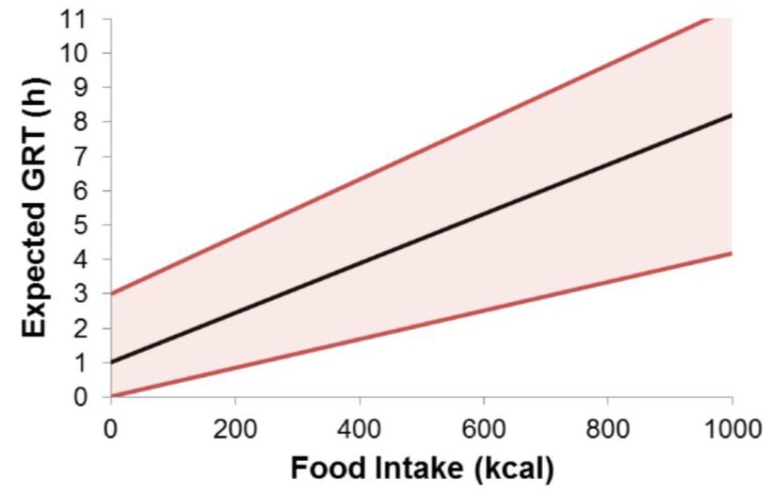
Expected maximum and minimum gastric residence times (GRTs) for larger, non-disintegrating objects according to the above stated rule of thumb (solid, red lines). The solid, black line represents the linear relationship demonstrated by Waterman [12].

**Figure 16 pharmaceutics-11-00416-f016:**
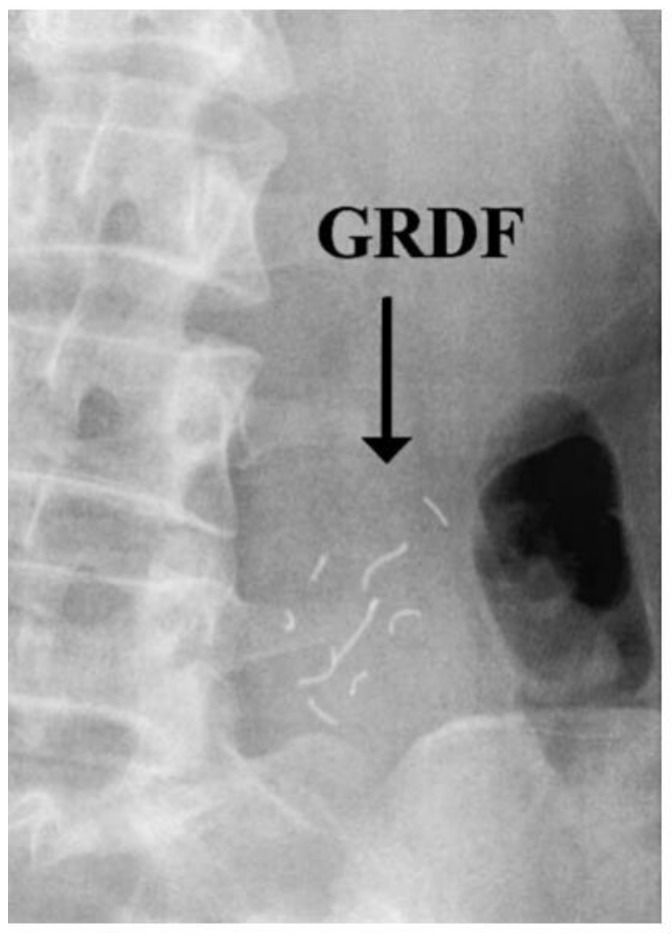
X-ray image of the unfolding system proposed by Klausner et al. and reprinted from Reference [131] with permission from Springer Nature.

**Figure 17 pharmaceutics-11-00416-f017:**
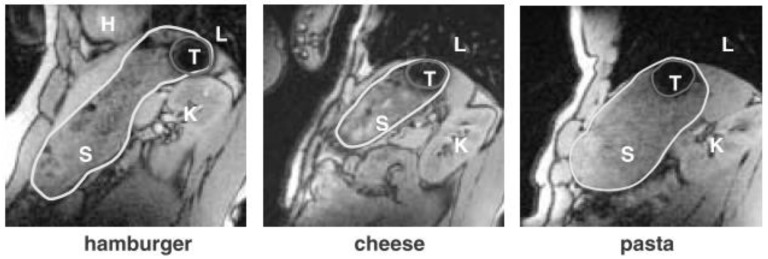
Magnetic resonance imaging of an iron-oxide-labeled floating tablet (T) in the stomach (S) on top of different meals. (H) heart, (K) kidney, (L) lung. Reprinted from Reference [42] with permission from John Wiley and Sons.

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
