# Peer review of "In Vitro and In Vivo Test Methods for the Evaluation of Gastroretentive Dosage Forms"

_pharmaceutics, 2019, doi:10.3390/pharmaceutics11080416_

Round 1
Reviewer 1 Report
The manuscript provides comprehensive review and critical thinking on the current state-of-the-art related to development and evaluation of gastroretentive dosage forms. It is well written and will complement the existing literature in the field.
The authors may consider some minor revisons:
- schematic presentation of different concepts towards gastroretention (Figures 4-7) could be merged into one Figure;
- Figures 14, 16 and 20 might be excluded from the revised manuscript;
line 75 – reconsider the use of term “chapter”
line 438 – reconsider the use of term “fatty” media
line 639 – adjective to is missing (“… in order to assure …”)
Author Response
The manuscript provides comprehensive review and critical thinking on the current state-of-the-art related to development and evaluation of gastroretentive dosage forms. It is well written and will complement the existing literature in the field.
The authors may consider some minor revisons:
- schematic presentation of different concepts towards gastroretention (Figures 4-7) could be merged into one Figure;
Thanks for your comment. We agree with the reviewer and merged the Figures into one.
- Figures 14, 16 and 20 might be excluded from the revised manuscript;
We fully agree that the mentioned figures can be excluded from the manuscript as they provide rather little support for the text. We therefore deleted the figures as recommended.
line 75 – reconsider the use of term “chapter”
Thanks. We changed the wording from “chapter” to “section” at the mentioned line and furthermore in line 117.
line 438 – reconsider the use of term “fatty” media
We fully agree. We revised this part and changed the wording to “more complex media (e.g. emulsions)”.
line 639 – adjective to is missing (“… in order to assure …”)
Thank you. We corrected the mistake.
Reviewer 2 Report
The manuscript titled “In vitro and in vivo test methods for the evaluation of gastroretentive dosage forms” is an interesting review article highlighting various methods for gastroretentive dosage forms evaluation. Generally, the article is well-written and I suggest following remarks for improvement.
Line 14, please can the author make In vitro and in vivo italic, and make these changes throughout the manuscript.
Line 17, If important aspects……………….. cannot be assessed correctly, this sentence is slight confused please can author think about rewriting.
Line 175, Figures 4-7 have been placed at the start of the section which I think unusual, please can I ask authors to use the Figures in the text or after the text.
Also, it is advised to develop some Tables to highlight section 3 (Formulation strategies) and section 4 (Characterization of gastroretentive dosage forms).
Author Response
The manuscript titled “In vitro and in vivo test methods for the evaluation of gastroretentive dosage forms” is an interesting review article highlighting various methods for gastroretentive dosage forms evaluation. Generally, the article is well-written and I suggest following remarks for improvement.
Line 14, please can the author make In vitro and in vivo italic, and make these changes throughout the manuscript.
Thanks for your comment. We carefully revised the entire manuscript and corrected the mistakes as recommended.
Line 17, If important aspects……………….. cannot be assessed correctly, this sentence is slight confused please can author think about rewriting.
Thanks. We fully agree that this sentence is too complex for the use in the abstract. We split it in two so that the understanding may hopefully be increased.
Line 175, Figures 4-7 have been placed at the start of the section which I think unusual, please can I ask authors to use the Figures in the text or after the text.
We carefully revised this part of the manuscript and merged Figures 4-7 into one figure. This was placed in the text as recommended by the reviewer.
Also, it is advised to develop some Tables to highlight section 3 (Formulation strategies) and section 4 (Characterization of gastroretentive dosage forms).
We carefully revised these sections but could not think of a proper table to express or highlight important aspects, which at the same time provide further information that is not explicitly stated in the text. If a table is not absolutely mandatory for the reviewer, we would like to keep the manuscript in its present form.